# Neural Execution Engines: Learning to Execute Subroutines

**Yujun Yan**[*]
The University of Michigan
yujunyan@umich.edu

**Kevin Swersky**
Google Research
kswersky@google.com

**Danai Koutra**
The University of Michigan
dkoutra@umich.edu

**Parthasarathy Ranganathan, Milad Hashemi**
Google Research
{parthas, miladh}@google.com

## Abstract

A significant effort has been made to train neural networks that replicate algorithmic reasoning, but they often fail to learn the abstract concepts underlying these algorithms. This is evidenced by their inability to generalize to data distributions that are outside of their restricted training sets, namely larger inputs and unseen data. We study these generalization issues at the level of numerical subroutines that comprise common algorithms like sorting, shortest paths, and minimum spanning trees. First, we observe that transformer-based sequence-to-sequence models can learn subroutines like sorting a list of numbers, but their performance rapidly degrades as the length of lists grows beyond those found in the training set. We demonstrate that this is due to attention weights that lose fidelity with longer sequences, particularly when the input numbers are numerically similar. To address the issue, we propose a learned conditional masking mechanism, which enables the model to strongly generalize far outside of its training range with near-perfect accuracy on a variety of algorithms. Second, to generalize to unseen data, we show that encoding numbers with a binary representation leads to embeddings with rich structure once trained on downstream tasks like addition or multiplication. This allows the embedding to handle missing data by faithfully interpolating numbers not seen during training.

## 1  Introduction

Neural networks have become the preferred model for pattern recognition and prediction in perceptual tasks and natural language processing [13, 4] thanks to their flexibility and their ability to learn complex solutions. Recently, researchers have turned their attention towards imbuing neural networks with the capability to perform *algorithmic reasoning*, thereby allowing them to go beyond pattern recognition and logically solve more complex problems [9, 12, 10, 14]. These are often inspired by concepts in conventional computer systems (e.g., pointers [29], external memory [22, 9]).

Unlike perceptual tasks, where the model is only expected to perform well on a specific distribution from which the training set is drawn, in algorithmic reasoning the goal is to learn a robust solution that performs the task *regardless* of the input distribution. This ability to generalize to *arbitrary* input distributions—as opposed to unseen instances from a fixed data distribution—distinguishes the concept of *strong generalization* from ordinary generalization. To date, neural networks still have difficulty learning algorithmic tasks with strong generalization [28, 8, 12].

---

[*]Work completed during an internship at Google.

In this work, we study this problem by learning to imitate the composable subroutines that form the basis of common algorithms, namely selection sort, merge sort, Dijkstra's algorithm for shortest paths, and Prim's algorithm to find a minimum spanning tree. We choose to focus on subroutine imitation as: (1) it is a natural mechanism that is reminiscent of how human developers decompose problems (e.g., developers implement very different subroutines for merge sort vs. selection sort), (2) it supports introspection to understand how the network may fail to strongly generalize, and (3) it allows for providing additional supervision to the neural network if necessary (inputs, outputs, and intermediate state).

By testing a powerful sequence-to-sequence transformer model [26] within this context, we show that while it is able to learn subroutines for a given data distribution, it fails to strongly generalize as the test distribution deviates from the training distribution. Subroutines often operate on strict subsets of data, and further analysis of this failure case reveals that transformers have difficulty separating "what" to compute from "where" to compute, manifesting in attention weights whose entropy increases over longer sequence lengths than those seen in training. This, in turn, results in misprediction and compounding errors.

Our solution to this problem is to leverage the transformer mask. First, we have the transformer predict both a value and a pointer. These are used as the current output of the subroutine, and the pointer is also used as an input to a learned conditional masking mechanism that updates the encoder mask for subsequent computation. We call the resulting architecture a Neural Execution Engine (NEE), and show that NEEs achieve near-perfect generalization over a significantly larger range of test values than existing models. We also find that a NEE that is trained on one subroutine (e.g., comparison) can be used in a variety of algorithms (e.g., Dijkstra, Prim) *as-is without retraining*.

Another essential component of algorithmic reasoning is representing and manipulating numbers [30]. To achieve strong generalization, the employed number system must work over large ranges and generalize outside of its training domain (as it is intractable to train the network on *all* integers). In this work, we leverage binary numbers, as binary is a hierarchical representation that expands exponentially with the length of the bit string (e.g., 8-bit binary strings represent exponentially more data than 7-bit binary strings), thus making it possible to train and test on significantly larger number ranges compared to prior work [8, 12]. We demonstrate that the binary embeddings trained on downstream tasks (e.g., addition, multiplication) lead to well-structured and interpretable representations with natural interpolation capabilities.

## 2 Background

### 2.1 Transformers and Graph Attention Networks

Transformers are a family of models that represent the current state-of-the-art in sequence learning [26, 4, 19]. Given input token sequences $\mathbf{x}_1, \mathbf{x}_2, \ldots, \mathbf{x}_{L_1} \in \mathcal{X}$ and output token sequences $\mathbf{y}_1, \mathbf{y}_2, \ldots, \mathbf{y}_{L_2} \in \mathcal{Y}$, where $\mathbf{x}_i, \mathbf{y}_j \in \mathbb{Z}^+$, a transformer learns a mapping $\mathcal{X} \to \mathcal{Y}$. First, the tokens are individually embedded to form $\hat{\mathbf{x}}_i, \hat{\mathbf{y}}_j \in \mathbb{R}^d$. The main module of the transformer architecture is the self-attention layer, which allows each element of the sequence to concentrate on a subset of the other elements.[2] Self-attention layers are followed by a point-wise feed-forward neural network layer, forming a self-attention block. These blocks are composed to form the encoder and decoder of the transformer, with the outputs of the encoder being used as queries and keys for the decoder. More details can be found in [26].

An important component for our purposes is the self-attention mask. This is used to prevent certain positions from propagating information to other positions. A mask, $\mathbf{b}$, is a binary vector where the value $b_i = 0$ indicates that the $i^{\text{th}}$ input should be considered, and $b_i = 1$ indicates that the $i^{\text{th}}$ input should be ignored. The vector $\mathbf{b}$ is broadcast to zero out attention weights of ignored input numbers. Typically this is used for decoding, to ensure that the model can only condition on past outputs during sequential generation. Graph Attention Networks [27] are essentially transformers where the encoder mask reflects the structure of a given graph. In our case, we will consider masking in the encoder as an explicit way for the model to condition on the part of the sequence that it needs at a given point in its computation, creating a dynamic graph. We find that this focuses the attention of the transformer and is a critical component for achieving strong generalization.

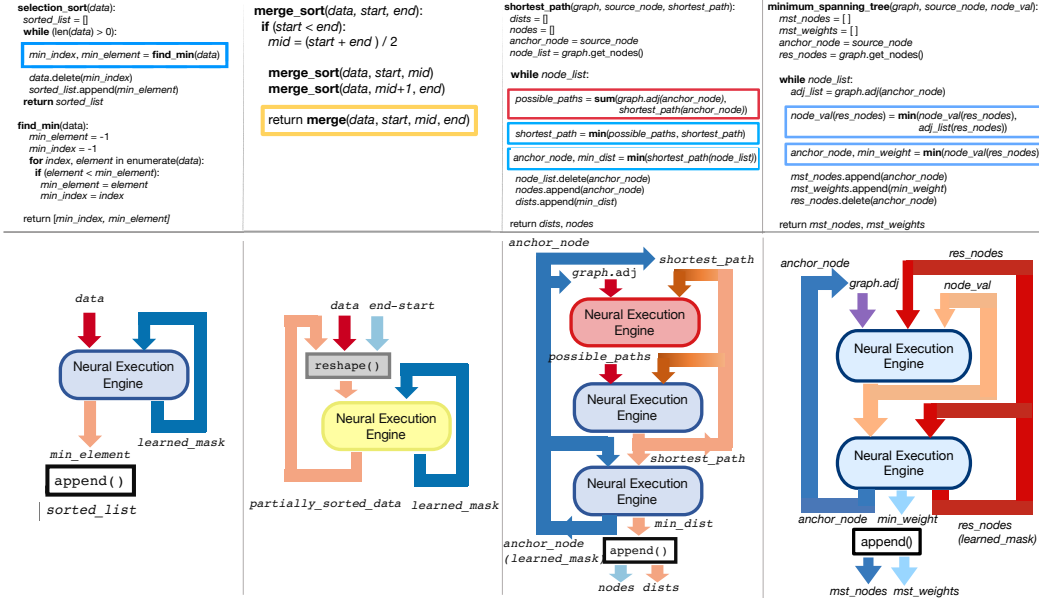

Figure 1: (Top) Pseudocode for four common algorithms: selection sort, merge sort, Dijkstra's algorithm for shortest paths, and Prim's algorithm for minimum spanning tree. Blue/red/yellow boxes highlight comparison, arithmetic, and difficult pointer manipulation subroutines respectively. (Bottom) Flow charts of the algorithms with NEE components implementing the subroutines.

## 2.2 Numerical Subroutines for Common Algorithms

We draw examples from different algorithmic categories to frame our exploration into the capability of neural networks to perform generalizable algorithmic reasoning. Figure 1 shows the pseudocode and subroutines for several commonly studied algorithms; specifically, selection sort, merge sort, Dijkstra's algorithm for shortest paths, and Prim's algorithm to find a minimum spanning tree. These algorithms contain a broad set of subroutines that we can classify into three categories:

- **Comparison subroutines** are those involving a comparison of two or more numbers.

- **Arithmetic subroutines** involve transforming numbers through arithmetic operations (we focus on addition in Figure 1, but explore multiplication later).

- **Pointer manipulation** requires using numerical values (pointers) to manipulate other data values in memory. One example is shown for merge sort, which requires merging two sorted lists. This could be trivially done by executing another sort on the concatenated list, however the aim is to take advantage of the fact that the two lists are sorted. This involves maintaining pointers into each list and advancing them only when the number they point to is selected.

## 2.3 Number Representations

Beyond subroutines, numerics are also critically important in teaching neural networks to learn algorithms. Neural networks generally use either categorical, one-hot, or integer number representations. Prior work has found that scalar numbers have difficulty representing large ranges [25] and that binary is a useful representation that generalizes well [12, 21]. We explore embeddings of binary numbers as a form of representation learning, analogous to word embeddings in language models [15], and show that they learn useful structure for algorithmic tasks.

# 3 Neural Execution Engines

A neural execution engine (NEE) is a transformer-based network that takes as input binary numbers and an encoding mask, and outputs either data values, a pointer, or both [3]. Here, we consider input and output data values to be $n$-bit binary vectors, or a sequence of such vectors, and the output pointer to be a one-hot vector of the length of the input sequence. The pointer is used to modify the mask the next time the NEE is invoked. A NEE is essentially a graph attention network [27] that can modify its own graph, resulting in a new decoding mechanism.

The NEE architecture is shown in Figure 2. It is a modification of the transformer architecture. Rather than directly mapping one sequence to another, a NEE takes an input sequence and mask indicating which elements are relevant for computation. It encodes these using a masked transformer encoder. The decoder takes in a single input, the zero vector, and runs the transformer decoder to output a binary vector corresponding to the output value. The last layer of attention in the last decoder block is used as a pointer to the next region of interest for computation. This and the original mask vector are fed into a final temporal convolution block that outputs a new mask. This is then applied in a recurrent fashion until there are no input elements left to process (according to the mask). In the remainder of this section, we go into more detail on the specific elements of the NEE.

**Conditional Masking** The encoder of a NEE takes as input both a set of values and a mask, which is used to force the encoder to ignore certain inputs. *We use the output pointer of the decoder to modify the mask for a subsequent call of the encoder.* In this way, the inputs represent a memory state, and the mask represents a set of pointers into the memory. A NEE effectively learns where to focus its attention for performing computation.

Learning an attention mask update is a challenging problem in general, as the mask updates themselves also need to strongly generalize. In many algorithms, including the ones considered here, the mask tends to change within a local neighbourhood around the point of interest (the element pointed to by the NEE). For example, in iterative algorithms, the network needs to attend to the next element to be processed which is often close to the last element that was processed.

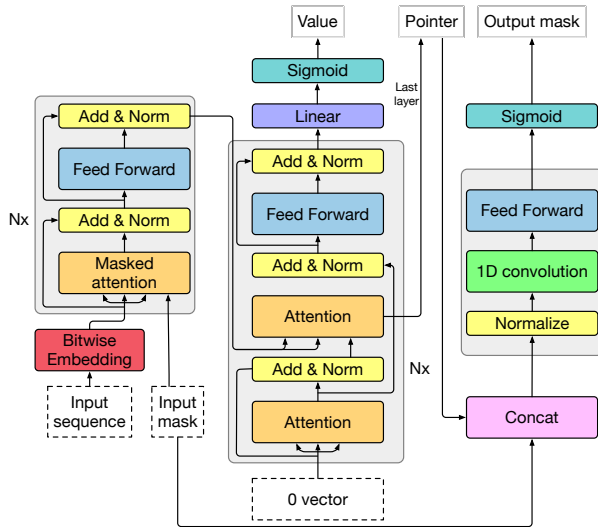

Figure 2: NEE architecture. The first step of decoding is equivalent to passing in a trainable decoder vector.

We therefore use a small temporal (1D) convolutional neural network (CNN), $T$. The CNN accepts as input the current mask vector $\mathbf{b}^I$ and the one-hot encoded output pointer $\mathbf{b}^P$ from the decoder. It outputs the next mask vector $\hat{\mathbf{b}}$. Mathematically, $\hat{\mathbf{b}} = \sigma(T(\mathbf{b}^I \parallel \mathbf{b}^P)) = \sigma(F(C(N(\mathbf{b}^I \parallel \mathbf{b}^P))))$, where $\parallel$ denotes concatenation, $\sigma$ denotes sigmoid function, $F$, $C$ and $N$ represent the point-wise feed-forward layer, 1D convolutional layer and feature-wise normalization layer, respectively. At inference, we simply choose the argmax of the pointer output head to produce $\mathbf{b}^P$.

The intuition behind this design choice is that through convolution, we enforce a position ordering to the input by exchanging information among the neighbourhoods. The convnet is shift invariant and therefore amenable to generalizing over long sequences. We also experimented with a transformer encoder-decoder, using an explicit positional encoding, however we found that this often fails due to the difficulty in dealing with unseen positions.

**Bitwise Embeddings** As input to a NEE, we embed binary vectors using a linear projection. This is equivalent to defining a learnable vector for each bit position, and then summing these vectors

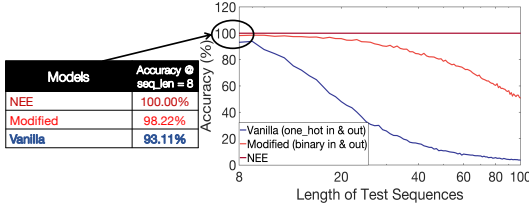

| Models | Accuracy @ seq_len = 8 |
|---|---|
| NEE | 100.00% |
| Modified | 98.22% |
| Vanilla | 93.11% |

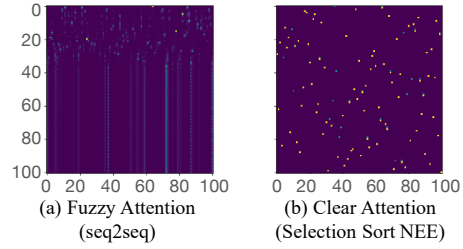

(a) Fuzzy Attention (seq2seq)

(b) Clear Attention (Selection Sort NEE)

Figure 3: Sorting performance of transformers trained on sequences of up to length 8.

Figure 4: Visualizing decoder attention weights. Attention is over each row. Transformer attention saturates as the output sequence length increases, while NEE maintains sharp attention.

elementwise, modulated by the value of their corresponding bit. That is, given an embedding vector $\mathbf{v}_i$ for each bit $i$, for an $n$-bit input vector $\mathbf{x}$, we would compute $\hat{\mathbf{x}} = \sum_{i=1}^{n} x_i \mathbf{v}_i$. For example, $\text{emb}(1001) = v_0 + v_3$.

Two important tokens for our purposes are *start s* and *end e*. These are commonly used in natural language data to denote the start and end of a sequence. We use $s$ as input to the decoder, and $e$ to denote the end of an input sequence. This allows us to train a NEE to learn to emit $e$ when it has completed an algorithm.

Additionally, we also use these symbols to define both 0 and $\infty$. These concepts are important for many algorithms, particularly for initialization. For addition, we require that $0 + x = x$ and $\infty + x = \infty$. As a more concrete example, in shortest path, the distance from the source node to other nodes in the graph can be denoted by $\infty$ since they're unexplored. We set $s = 0$ and train the model to learn an embedding vector for $e$ such that $e = \infty$. That is, the model will learn $e > x$ for all $x \neq e$ and that $e + x = e$.

# 4 Current Limitations of Sequence to Sequence Generalization

**Learning Selection Sort**    We first study how well a state-of-the-art transformer-based sequence to sequence model (Section 2.1) learns selection sort. Selection sort involves iteratively finding the minimum number in a list, removing that number from the list, and adding it to the end of the sorted list. We model selection sort using sequence to sequence learning [24] with input examples of unsorted sequences ($L \leq 8$ integers, each within the range $[0, 256)$) and output examples of the correctly sorted sequences. To imitate a sorting subroutine, we provide supervision on intermediate states: at each stage of the algorithm the transformer receives the unsorted input list, the partially sorted output list, and the target number. The numbers used as inputs and outputs to a vanilla transformer are one-hot encoded[4]. The decoder uses a greedy decoding strategy.

The performance of this vanilla transformer, evaluated as achieving an exact content and positional match to the correct output example, is shown in Figure 3. The vanilla transformer is able to learn to sort the test-length distribution (at 8 numbers) reasonably well, but performance rapidly degrades as the input data distribution shifts to longer sequences and by 100 integers, performance is under 10%.

One of the main issues we found is that the vanilla transformer has difficulty distinguishing close numbers (e.g., 1 vs. 2, 53 vs. 54)[5]. We make a number of small architectural modifications in order to boost its accuracy in this regime, including but not limited to using a binary representation for the inputs and outputs. We describe these modifications, and provide ablations in Appendix A.2.

As Figure 3 also shows, given our modifications to a vanilla transformer, sequence-to-sequence transformers are capable of learning this algorithm on sequences of length $\leq 8$ with a high degree of accuracy. However, the model still fails to generalize to longer sequences than those seen at training time, and performance sharply drops as the sequence length increases.

**Attention Fidelity**  To understand why performance degrades as the test sequences get longer, we plot the attention matrix of the last layer in the decoder (Figure 4a). During decoding, the transformer accurately attends to the first few numbers in the sequence (distinct dots in the chart) but the attention distribution becomes "fuzzy" as the number of decoding steps increases beyond 8 numbers, often resulting in the same number being repeatedly predicted.

Since the transformer had difficulty clearly attending to values beyond the training sequence length, we separate the supervision of *where* the computation needs to occur from *what* the computation is. *Where* the computation needs to occur is governed by the transformer mask. To avoid overly soft attention scores, we aim to restrict the locations in the unsorted sequence where the transformer could possibly attend in every iteration. This is accomplished by producing a conditional mask which learns to ignore the data elements that have already been appended to the `sorted_list` and feed that mask back into the transformer (shown on the bottom-left side of Figure 1). Put another way, we have encoded the current algorithmic state (the sorted vs. unsorted list elements) in the attention mask rather than the current decoder output.

This modification separates the control (which elements should be considered) from the computation itself (find the minimum value of the list). This allows the transformer to learn output logits of much larger magnitude, resulting in sharper attention, as shown in Figure 4b. Our experimental results consequently demonstrate strong generalization, sorting sequences of up to length 100 without error, as shown in Figure 3. Next, we evaluate this mechanism on a variety of other algorithms.

## 5 Experiments

In this section, we evaluate using a NEE to execute various algorithms, including selection sort and merge sort, as well as more complex graph algorithms like shortest path and minimum spanning tree.

### 5.1 Executing Subroutines

**Selection Sort**  Selection sort (described in Sec. 4) is translated to the NEE architecture in Figure 1. The NEE learns to find the minimum of the list, and learns to iteratively update the mask by setting the mask value of the location of the minimum to 1. We show the results for selection sort in Figure 3 and Table 1, the NEE is able to strongly generalize to inputs of length 100 with near-perfect accuracy.

Table 1: Performance of different tasks on variable sizes of test examples (trained with examples of size 8). *Two exceptions: accuracy for graphs of 92 nodes and 97 nodes are 99.99 and 99.98, respectively. We run the evaluation once, and minimally tune hyper-parameters.

| Accuracy \ Sizes | 25 | 50 | 75 | 100 |
|---|---|---|---|---|
| Selection sort | 100.00 | 100.00 | 100.00 | 100.00 |
| Merge sort | 100.00 | 100.00 | 100.00 | 100.00 |
| Shortest path | 100.00 | 100.00 | 100.00 | 100.00* |
| Minimum spanning tree | 100.00 | 100.00 | 100.00 | 100.00 |

**Merge Sort**  The code for one implementation of merge-sort is shown in Figure 1. It is broadly broken up into two subroutines, data decomposition (`merge_sort`) and an action (`merge`). Every call to `merge_sort` divides the list in half until there is one element left, which by definition is already sorted. Then, `merge` unrolls the recursive tree, combining every 2 elements (then every 4, 8, etc.) until the list is fully sorted. Recursive algorithms like merge sort generally consist of these two steps (the "recursive case" and the "base case").

We focus on the merge function, as it involves challenging pointer manipulation. For two sorted sequences that we would like to merge, we concatenate them and delimit them using the $e$ token: $[\text{seq}_1, e, \text{seq}_2, e]$. Each sequence has a pointer denoting the current number being considered, represented by setting that element to $0$ in the mask and all other elements in that sequence to $1$, e.g., $\mathbf{b}_{\text{init}} = [0\ 1\ 1\ 0\ 1\ 1]$ for two length-2 sequences delimited by $e$ tokens. The smaller of the two currently considered numbers is chosen as the next number in the merged sequence. The pointer for the chosen sequence is advanced by masking out the current element in the sequence and unmasking the next, and the subroutine repeats.

More concretely, the NEE in Figure 1 implements this computation. Every timestep, the model outputs the smallest number from the unmasked numbers and the two positions to be considered next. When the pointers both point to $e$, then the subroutine returns. Table 1 demonstrates that the NEE is

able to strongly generalize on merge sort over long sequences (up to length 100) while trained on sequences of length smaller or equal to 8.

**Composable Subroutines: Shortest Path**  While both merge sort and selection sort demonstrated that a NEE can compose the same subroutine repeatedly to sort a list with perfect accuracy, programs often compose multiple different subroutines to perform more complex operations. In this section, we study whether multiple NEEs can be composed to execute a more complicated algorithm.

To that end, we study a graph algorithm, Dijkstra's algorithm to find shortest paths, shown in Figure 1. The algorithm consists of four major steps:

(1) Initialization: set the distance from the source node to the other nodes to infinity, then append them into a queue structure for processing; (2) Compute newly found paths from the source node to all neighbours of the selected node; (3) Update path lengths if they are smaller than the stored lengths; (4) Select the node with the smallest distance to the source node and remove it from the queue. The algorithm repeats steps (2)–(4) as long as there are elements in the queue.

Computing Dijkstra's algorithm requires the NEEs to learn the three corresponding subroutines (Figure 1). Finding the minimum between the `possible_paths` and `shortest_path` as well as the minimum current `shortest_path` can be accomplished through the NEE trained to accomplish the same goal for sorting. The new challenge is to learn a numerical subroutine, addition. This process is described in detail in Section 5.2.

We compose pre-trained NEEs to perform Dijkstra's algorithm (Figure 1). The NEEs themselves strongly generalize on their respective subroutines, therefore they also strongly generalize when composed to execute Dijkstra's algorithm. This persists across a wide range of graph sizes. A step-by-step view is shown in the Appendix. The examples are Erdős-Rényi random graphs. We train on graphs with up to 8 nodes and test on graphs of up to 100 nodes, with 100 graphs evaluated at each size. Weights are randomly assigned within the allowed 8-bit number range. We evaluate the prediction accuracy on the final output (the shortest path of all nodes to the source nodes) and achieve 100% test accuracy with graph sizes up to 100 nodes (Table 1).

**Composable Subroutines: Minimum Spanning Tree**  As recent work has evaluated generalization on Prim's algorithm [28], we include it in our evaluation. This algorithm is shown in Figure 1: We compose pre-trained NEEs to compute the solution, training on graphs of 8 nodes and testing on graphs of up to 100 nodes. The graphs are Erdős-Rényi random graphs. We evaluate the prediction accuracy on the whole set, which means the prediction is correct if and only if the whole set predicted is a minimum spanning tree. Table 1 shows that we achieve strong generalization on graphs of up to 100 nodes, whereas [28] sees accuracy drop substantially at this scale. We also test on other graph types (including those from [28]) and perform well. Details are provided in Appendix A.3.

## 5.2 Number representations

**Learning Arithmetic**  A core component of many algorithms, is simple addition. While neural networks internally perform addition, our goal here is to see if NEEs can learn an internal number system using binary representations. This would allow it to gracefully handle missing data and can serve as a starting point towards more complex numerical reasoning. To gauge the relative difficulty of this versus other arithmetic tasks, we also train a model for multiplication.

The results are shown in Table 2. Training on the entire 8-bit number range (256 numbers) and testing on unseen *pairs* of numbers, the NEE achieves 100% accuracy. In addition to testing on unseen pairs, we test performance on *completely unseen numbers* by holding out random numbers during training. These results are also shown in Table 2, and the NEE demonstrates high performance even while training on 25% of the number range (64 numbers). This is a promising result as it suggests that we may be able to extend the framework to significantly larger bit vectors, where observing every

Table 2: NEE 8-bit addition performance.

| Training Numbers | 256 | 224 | 192 | 128 | 89 | 76 | 64 |
|---|---|---|---|---|---|---|---|
| **Accuracy%** | 100.00 | 100.00 | 100.00 | 100.00 | 100.00 | 99.00 | 96.53 |

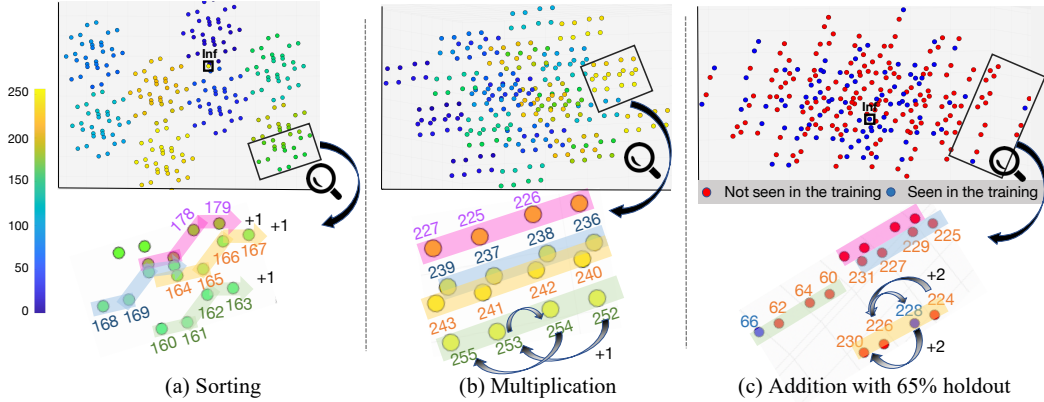

(a) Sorting         (b) Multiplication        (c) Addition with 65% holdout

Figure 5: 3D PCA visualization of learned bitwise embeddings for different numeric tasks. The embeddings exhibit regular, task-dependent structure, even when most numbers have not been seen in training (c).

number in training is intractable. In the case of multiplication, we train on 12-bit numbers and also observe 100% accuracy.

To understand the number system that the NEE has learned, we visualize the structure of the learned embeddings using a 3-dimensional PCA projection, and compare the embeddings learned from sorting, multiplication, and addition, shown in Figure 5 (a), (b), and (c) respectively. For the addition visualization, we show the embeddings with 65% of the numbers held out during training. In Figure 5 (a) and (b), each node is colored based on the number it represents; in Figure 5 (c), held-out numbers are marked red. We find that a highly structured number system has been learned for each task. The multiplication and addition embeddings consist of multiple lines that exhibit human-interpretable patterns (shown with arrows in Figure 5 (b) and (c)). The sorting embeddings exhibit many small clusters, and the numbers placed in a "Z" curve increase by 1 (shown with arrows in Figure 5 (a)). On held out numbers for the addition task, NEE places the embeddings of the unseen numbers in their logical position, allowing for accurate interpolation. More detailed visualizations are provided in Appendix A.4.

# 6 Related Work

**Learning subroutines**    Inspired by computing architectures, there have been a number of proposals for neural networks that attempt to learn complex algorithms purely from *weak supervision*, i.e., input/output pairs [9, 10, 12, 14, 29]. Theoretically, these are able to represent any computable function, though practically they have trouble with sequence lengths longer than those seen in training and do not strongly generalize. Unlike these networks that are typically trained on scalar data values in limited ranges, focus purely on pointer arithmetic, or contain non-learnable subroutines, we train on significantly larger (8-bit) number ranges, and demonstrate strong generalization in a wide variety of algorithmic tasks.

Recent work on neural execution [28] explicitly models intermediate execution states (*strong supervision*) in order to learn *graph* algorithms. They also find that the entropy of attention weights plays a significant role in generalization, and address the problem by using max aggregation and entropy penalties [28]. Despite this solution, a drop in performance is observed over larger graphs, including with Prim's algorithm. On the other hand, in this work, we demonstrate strong generalization on Prim's algorithm on much larger graphs than those used in training (Section 5). NEE has the added benefit that it does not require additional heuristics to learn a low-entropy mask—it naturally arises from conditional masking.

Work in neural program synthesis [16, 18, 5, 1, 6]—which uses neural networks with the goal of generating and finding a "correct" program such that it will generalize beyond the training distribution—has also employed *strong supervision* in the form of execution traces [20, 21, 2]. For instance, [2] uses execution traces with tail recursion (where the subroutines call themselves).

Though [2] shows that recursion leads to improved generalization, their model relies on predefined non-learnable operations like SHIFT & SWAP.

The computer architecture community has also explored using neural networks to execute approximate portions of algorithms, as there could be execution speed and efficiency advantages [7]. Unlike traditional microprocessors, neural networks and the NEE are a non-von Neumann computing architecture [30]. Increasing the size of our learned subroutines could allow neural networks and learned algorithms to replace or augment general purpose CPUs on specific tasks.

**Learning arithmetic**   Several works have used neural networks to learn number systems for performing arithmetic, though generally on small number ranges [3]. For example, [17] directly embeds integers in the range $[-10, 10]$ as vectors and trains these, along with matrices representing relationships between objects. [23] expands on this idea, modeling objects as matrices so that relationships can equivalently be treated as objects, allowing the system to learn higher-order relationships. [25] explores the (poor) generalization capability of neural networks on scalar-values inputs outside of their training range, and develops new architectures that are better suited to scalar arithmetic, improving extrapolation.

Several papers have used neural networks to learn binary arithmetic with some success [11, 12]. [8] develops a custom architecture that is tested on performing arithmetic, but trains on symbols in the range of $[1, 12]$ and does not demonstrate strong generalization. Also, recent work has shown that graph neural networks are capable of learning from 64-bit binary memory states provided execution traces of assembly code, and observes that this representation numerically generalizes better than one-hot or categorical representations [21]. Going beyond this, we *directly* explore computation with binary numbers, and the resultant structure of the learned representations.

# 7    Conclusion

We propose neural execution engines (NEEs), which leverage a learned mask to imitate the functionality of larger algorithms. We demonstrate that while state-of-the-art sequence models (transformers) fail to strongly generalize on tasks like sorting, imitating the smaller subroutines that compose to form a larger algorithm allows NEEs to strongly generalize across a variety of tasks and number ranges. There are many natural extensions within and outside of algorithmic reasoning. For example, one could use reinforcement learning to replace imitation learning, and learn to increase the efficiency of known algorithms, or link the generation of NEE-like models to source code. Growing the sizes of the subroutines that a NEE learns could allow neural networks to supplant general purpose machines for execution efficiency, since general-purpose machines require individual sequentially encoded instructions [30]. Additionally, the concept of strong generalization allows us to reduce the size of training datasets, as a network trained on shorter sequences or small graphs is able to extrapolate to much longer sequences or larger graphs, thereby increasing training efficiency. We also find the link between learned attention masks and strong generalization as an interesting direction for other areas, like natural language processing.

# 8    Broader Impact of this Work

This work is a very incremental step in a much broader initiative towards neural networks that can perform algorithmic reasoning. Neural networks are currently very powerful tools for perceptual reasoning, and being able to combine this with algorithmic reasoning in a single unified system could form the foundation for the next generation of AI systems. True strong generalization has a number of advantages: strongly generalizing systems are inherently more reliable. They would not be subject to issues of data imbalance, adversarial examples, or domain shift. This could be especially useful in many important domains like medicine. Strong generalization can also reduce the size of datasets required to learn tasks, thereby also providing environmental savings by reducing the carbon footprint of running large-scale workloads. However, strong generalization could be more susceptible to inheriting the biases of the algorithms on which they are based. If the underlying algorithm is based on incorrect assumptions, or limited information, then strong generalization will simply reflect this, rather than correct it.

## Acknowledgments and Disclosure of Funding

We thank Danny Tarlow and the anonymous NeurIPS reviewers for their insightful feedback. This work was supported by a Google Research internship and a Google Faculty Research Award.

## Footnotes

[2] We do not use positional encodings (which we found to hurt performance) and use single-headed attention.

[3]Code for this paper can be found at https://github.com/Yujun-Yan/Neural-Execution-Engines

[4]We also experimented with one-hot 256-dimensional outputs for other approaches used in the paper with similar results. See the supplementary material.

[5]Throughout this work, the preponderance of errors are regenerated numbers that are off by small differences. Therefore, our test distribution consists of 60% random numbers and 40% numbers with small differences

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
