[Supplementary Material]

# A Appendix

## A.1 Hyperparameters

8-bit binary numbers are used in all tasks except the multiplication task, where 12-bit binary numbers are used. For sorting, we found it sufficient to use bitwise embeddings of dimension $d = 16$. For more difficult tasks like addition and multiplication, we found it necessary to increase the dimension to $d = 24$ and $d = 28$, respectively. We used no positional encoding for the sorting tasks, and single-headed attention for all tasks. The remaining NEE hyperparameters, aside from the changes described next, were set to their defaults.

Table 3: Architectural and training hyperparameters

| Hyperparameters | Value |
|---|---|
| Number of encoder (decoder) layers | 6 |
| Number of layers in the feed forward network | 2 |
| Number of hidden units in the feed forward network | 128 |
| Mask filter size | 3 |
| Mask number of filters | 16 |
| Ratio of residual connection | 1.5 |
| Dropout rate | 0.1 |
| Optimizer | Adam |
| Warm-up steps | 4000 |
| Learning rate | $\sqrt{d} \cdot \min(\sqrt{t},\ t \cdot 4000^{-1.5})$ |

## A.2 Sorting ablations

In this section, we extensively study the impact of different factors on the generalization ability of the transformer model. Specifically, we focus on attention mask supervision, encoding schemes and various architectural changes.

Unless otherwise specified, the task performed in this section is selection sort (Section 4). The models are trained on 20000 sequences of lengths no longer than 8 and tested on 100 sequences of various lengths. The numbers are integers in the range $[0, 256)$ and we include an end token into the number system. The test data consists of $60\%$ examples drawn uniformly, and $40\%$ drawn from a more difficult distribution, where the numbers are closer in value. The accuracy is measured by the percentage of correctly predicted numbers (correct position and values).

### A.2.1 Impact of supervision on attention masks

| Models | Accuracy @ seq_len = 8 |
|---|---|
| NEE | 100.00% |
| Vanilla (binary) | 98.11% |
| Vanilla (binary + sup) | 97.89% |

Figure 6: Sorting performances of the transformers w/o mask supervision

Figure 6 shows the sorting performance of the transformers w/o mask supervision. It can be seen that providing supervision on the mask hurts generalization ability during test, as the model can overfit on the attention mask during training.

Figure 7: Comparisons of one_hot and binary encoding schemes

## A.2.2 Impact of different encoding schemes

Figure 7 shows sorting performances with different encoding schemes. In general, vanilla transformers with binary encoded output perform better.

## A.2.3 Impact of different architectural changes

In Figure 3, we show the performance of a modified transformer model in a sequence-to-sequence setup (Section 4). In this section, we will elaborate on the modifications we made to the transformer, and how those modifications affect the generalization performance. We illustrate this by evaluating the results of selection sort.

We study 3 different data distributions: the first is where we train on uniformly random sequences with tokens in $[0, 255] \cup \{e\}$. The second is a mixed setting, where $60\%$ of the examples are drawn uniformly, and $40\%$ are drawn from a more difficult distribution, where the numbers are closer in value. The third is the most difficult setting, where all sequences have numbers that are close to each other in value.

(a) Vanilla encoder          (b) Modified encoder

Figure 8: Baseline transformer (a) and modified transformer (b).

We ablate specific architectural changes in these settings. The original and modified encoder are represented visually in in Figure 8. Specifically, the architectural choices we test are as follows and the ones applied in the modified transformer are checked (in that they provide a net benefit):

- $C1$: Scaling up the strength of the residual connections by a factor of 1.5 instead of 1. (✓)
- $C2$: Using an MLP-based attention module [1] instead of using the standard scaled dot product attention. (✓)
- $C3$: Symmetrizing the MLP-based attention by flipping the order of the inputs and averaging the resulting logit values. (✓)
- $C4$: Sharing the projection layer between the query, key, and value in the attention mechanism. (✓)

- $C5$: Using a binary encoding of input values instead of using a one-hot encoding of the input values. (✓)
- $C6$: Using a binary encoding as the input, but without any linear embedding.

We use the following conventions to refer to different transformer variants: all_mod stands for applying all the checked modifications, vanilla stands for the original transformer model with one-hot encoded input and "+" and "-" stand for choosing the corresponding modifications or keeping the original structures, respectively.

The test accuracy on sequences of length 8 in the *mixed* setting, is shown in Table 4. We can see that the architectural changes help improve performance on these sequences up to near-perfect accuracy.

Table 4: Seq2Seq performance for transformer variants at training length of 8 on mixed test sets.

| Models | Accuracy @ seq_len = 8 |
| --- | --- |
| all_mod | 99.00% |
| all_mod-C1 | 95.89% |
| all_mod-C2 | 97.56% |
| all_mod-C3 | 98.33% |
| all_mod-C4 | 98.44% |
| all_mod-C5 | 89.56% |
| all_mod+C6 | 84.78% |
| vanilla | 93.11% |
| vanilla+C5 | 96.67% |
| vanilla+C6 | 77.11% |

In Figure 9, we show the strong generalization performance of the different architectures. While some changes are able to improve performance in this regime, the performance ultimately drops steeply as the length of the test sequence increases. This is consistent across all test scenarios and suggests that standard modifications on the transformer architecture are *unlikely* to prevent attention weights from losing sharpness with longer sequences (Fig. 3).

Here we list out some random and hard examples as well as the corresponding output (containing some errors) from the vanilla transformer (each number has an independent embedding), which is commonly used in natural language models. The symbol e represents the end token. It can be seen that the model makes more mistakes (in bold and italics) with hard examples.

Random examples:
| 100 | 62 | 114 | 66 | 241 | 1 | 63 | 237 | e |
| 181 | 52 | 71 | 254 | 246 | 145 | 118 | 28 | e |

Output from vanilla:
| 1 | 62 | 63 | 66 | 100 | 114 | 237 | *53* | e |
| 28 | 52 | 71 | 118 | 145 | 181 | 246 | 254 | e |

Hard examples:
| 132 | 126 | 131 | 129 | 127 | 130 | 128 | 125 | e |
| 238 | 239 | 241 | 240 | 243 | 237 | 242 | 244 | e |

Output from vanilla:
| 125 | 126 | 127 | 128 | 129 | 130 | *132* | *e* | e |
| 237 | 238 | 240 | *244* | *243* | *e* | *237* | *242* | e |

Figure 9: Seq2Seq strong generalization performance on (a) mixed test sets, where test sets consist of 60% uniformly random examples and 40% hard examples where the numbers are close to each other. (b) uniformly random test sets. (c) hard test sets, where test sets consist of 100% hard examples where the numbers are close to each other. All models trained on sequences $\leq 8$ and tested up to length 100. Vanilla corresponds to the original transformer, with bitwise embeddings and all_modifications_excp_[change] means all modifications except a certain change.

Next, we will show the performance of transformer variants with a binary encoded output. From Figure 10, we can see that among all the models with binary encoded output, all_mod performs the best, which is consistent with the result obtained from one-hot encoded output models. Though all_mod with one-hot output outperforms all models with binary output, models with binary output use fewer parameters and scale better to the input data range.

Figure 10: Performances of transformer models (binary output) in mixed test sets

## A.3 Graph algorithms tested on different graph types

Prior work [6] has shown that performance on graph algorithms may depend on different types of graphs. For comparison, we further explore NEE performance on graph algorithms (Dijkstra and Prim) and we consider two scenarios: (1) Training NEEs with traces from selection sort (and addition) (2) Training NEEs with traces from corresponding graph algorithms and using Erdős-Rényi random graphs as training graphs. For both scenarios, we use 20000 training sequences/graphs of size 8 and 2000 validation sequences/graphs and test on 100 graphs of the following types with various sizes:

- Erdős-Rényi random graphs [3]: each pair of nodes has probability $p$ to form an edge, we use $p$ uniformly sampled from $[0, 1]$.

- Newman–Watts–Strogatz random graphs [4]: First create a ring of $n$ nodes, where each node is connected to its $k$ nearest neighbors (or $k - 1$ neighbors if $k$ is odd). Then for each edge $(u, v)$ in the original ring, add a new edge $(u, w)$ with probability $p$. We choose $2 \leq k \leq 5$ and $p \in [0, 1]$.

- D-regular random graphs: every node is connected to other $d$ nodes ($nd$ needs to be even and $2 \leq d \leq n$).

- Barabási–Albert random graphs [2]: A graph of $n$ nodes is grown by attaching new nodes each with $m$ edges that are preferentially attached to existing nodes with high degree. We choose $2 \leq m \leq 5$.

We assign random weights to the graphs such that they do not overflow the current number system (integers 0-255). Based on the findings in Section A.2 that close numbers are hard to identify, thus in the training data, $50\%$ ($20\%$) are hard examples (weights are very close) when training shortest paths (minimum spanning tree). All the training graphs are Erdős-Rényi random graphs while in the test graphs, every graph type contributes to 25 graph samples.

Table 5: Performance of graph algorithms with mixed graph types. The accuracy of Dijkstra's shortest paths is evaluated on the portion of correctly predicted shortest paths from all the other nodes to the source node. The accuracy of Prim's minimum spanning tree is evaluated on whether the predicted node sequence forms a minimum spanning tree and the corresponding edge weights are correct. Training with scenario 1(2) is labeled with $S_1(S_2)$ in parentheses.

| Accuracy \ Sizes | 25 | 50 | 75 | 100 |
|---|---|---|---|---|
| **Shortest path** ($S_1$) | 100.00 | 100.00 | 100.00 | 100.00 |
| **Minimum spanning tree** ($S_1$) | 100.00 | 100.00 | 100.00 | 100.00 |
| **Shortest path** ($S_2$) | 100.00 | 100.00 | 100.00 | 99.91 |
| **Minimum spanning tree** ($S_2$) | 100.00 | 99.00 | 93.00 | 92.00 |

From Table 5, we can see that NEEs are robust to different graph types and can achieve high performance when test graph sizes are much larger than the training graphs or the distributions of training and test graphs are different. Using Erdős-Rényi random graphs as training data ($S_2$), we observe a slight drop in performance due to distribution shift from the training data to the test data ([11]). This drop in performance does not occur using our subroutines trained to strong generalization ($S_1$).

Figure 11: Histograms of data used in MST ($S_2$)

### A.4 Detailed visualization of learned number embeddings

In Figure 12 we show more detailed visualizations of the learned bitwise embeddings. These are 3-dimensional PCA projections of the full embedding matrix, capturing approximately 75% of the total variance. The main takeaway is that the network is able to learn a coherent number system with a great deal of structure, and that this structure varies depending on the specific task of the network. This is reminiscent of [5], where linear embeddings learned the correct structure to solve a simple modular arithmetic task. Also, the network learns to embed infinity, outside of this structure.

Future work will also investigate the resulting embedding from a NEE that performs multiple or more complex tasks.

(a) Sorting Embedding

(b) Addition Embedding

(c) Generalization on Addition (numbers randomly held out of training colored red)

(d) Multiplication Embedding (250 numbers are shown)

Figure 12: 3-dimensional PCA projections of learned bitwise embeddings for (a) sorting, (b) addition, and (c) addition with 65% of the numbers withheld from training, (d) multiplication