[Reviews · NeurIPS 2020]

Review 1

Summary and Contributions: The paper addresses the task of learning to execute algorithmic reasoning using Transformer-based neural networks. The authors observe that transformer-based models can learn subroutines like sorting a list of numbers, but these models cannot generalize to larger lists of numbers. The authors analyze the reason for this lack of generalization ability and propose a simple conditional masking mechanism that allows the models to generalize to longer inputs significantly better. This is implemented in NEE - neural execution engine. The idea is that each transformer block masks some of its elements in the self-attention mechanism, and also predicts a pointer that allows modifying the mask of the next layer. The authors also propose to represent numbers using their binary representation for algorithmic tasks that require manipulating numbers.

Strengths: + The proposed model shows very good generalization abilities, in inputs that are much longer than the inputs the model was trained on. +The authors perform an evaluation on various numerical and graph-based tasks: selection sort, merge sort, shortest path, and minimum spanning tree.

Weaknesses: - There is a fundamental problem with the strong supervision that is required in this work. Besides the nice ability of the model to learn its labels, this dependency on the strong supervision makes the NEE model inapplicable for any real task. Since the NEE model required strong supervision at every layer - at best, it just learns to mimic the underlying algorithm (i.e., shortest path, selection sort). Therefore, we need to have the underlying algorithm to train the model. But if we have the underlying algorithm, we don't need a learning model to learn that. If the model could learn the algorithm only from examples (weaker supervision) - that would be useful in cases that we don't know the algorithm. If the model could learn a statistical algorithm (i.e., an algorithm that doesn't have a closed solution, like translation / ranking / recommendation) - that would be useful as well. We could encode all algorithms in history into neural networks and see how neural networks learn them - but would it be useful? Does it advance at toward an (even slightly) better understanding of neural networks? - The paper seems to be very related to [29], without a discussion of the differences, and without an empirical comparison. The authors indirectly compare their approach with [29] by saying that [29] did not generalize well to longer graphs in the "minimum spanning tree" task, while NEE does generalize well. However, this comparison was not performed while all other factors being equal, and the cause might be some other artifact such as a training detail, the binary representation, or the exact choice of data. - The authors discuss the idea of representing numbers using their binary representations and motivate this idea. However, the paper does not include an ablation that justifies this decision directly.

Correctness: The claims and the method seem correct.

Clarity: The paper is well written and easy to follow. I did not understand the visualization of Figure 4(a), I am not sure what is the difference between the axes, and what insights should we take from this figure.

Relation to Prior Work: The paper seems very related to [29] - "Neural execution of graph algorithms" (Velickovic et al., ICLR'2020). Although the paper mentions [29] and discusses the similarities between them, the paper does not discuss the *differences* between them. Why does NEE perform better than [29] in generalizing to longer sequences? What is the key ingredient compared to [29]? As stated above, the empirical comparison between this paper and [29] is indirect and not performed in the same settings and the same data.

Reproducibility: Yes

Additional Feedback: The authors submitted their code which is good for facilitating future research. Additional minor comments: * Some fonts in Figure 1 are unreadable * Figure 5 is very nice visually, but does not add new information or insight - isn't it expected that close numbers will be close in the PCA visualization? ==================================== Post-discussion ==================================== I think that the rhetoric arguments of this paper need to be phrased differently. For example, from the authors' response: "Generalization and strong generalization on out-of-distribution data (co-variate shift) are fundamentally unsolved problems in both machine learning" This is true, but generalization in the classical sense of machine learning is *not* what this paper addresses. I think that the paper addresses the question of: "whether neural models can learn algorithmic subroutines" and *not*: "whether neural models can achieve generalization" (or, as phrased in the authors' response: "Our primary objective with this paper is to establish whether strong generalization can be achieved") Because if we know the underlying algorithm/subroutines and the neural model learns to mimic it - this is not generalization, because the underlying algorithm is a perfect generalized by definition. So my point is that the authors' research question of "whether strong generalization can be achieved" is trivial: is, of course it can be achieved, if your model is/mimics the algorithm. I overall think that this is an interesting paper and (one of) the first to use transformers and pointerst to address algorithmic problems. Thus, I am keeping my score of 6. * Minor - I commented before that Figure 5 doesn't add any new information (and is thus redundant). Figure-R1 suffers again from the same problem: very nice visually, but totally expected and unnecessary (and takes space).


Review 2

Summary and Contributions: This paper addresses the generalization problem by learning to imitate the composable subroutines. They propose a learned conditional masking mechanism, and construct a Neural Execution Engine (NEE). This paper is outside my areas of expertise so I must apologise for the brief review.

Strengths: 1.The overall presentation of the paper is relatively acceptable (but can be significantly enhanced). 2.More ablation study is welcome. 3.The analyses and discussions of attention fidelity are good.

Weaknesses: 1. The motivation of the paper is not clear. Please state the motivation for each improvement by point-to-point matching. 2. In Table 1, the performance of different tasks with different parameters is perfect. Is this independent of parameters or tasks? Detailed discussion is required. 3. I perfer numerical analysis like Table 2 to visualization like Figure 5. So, I hope that the author can divide the clusters of different sizes for numerical analysis.

Correctness: Yes.

Clarity: Somewhat clear.

Relation to Prior Work: Yes.

Reproducibility: Yes

Additional Feedback:


Review 3

Summary and Contributions: This paper proposes to modify the Transformer architecture, so that it learns to execute sub-routines such as find_min, merge, sum and mul. Note that each Transformer architecture is specialized for one sub-routine, which is called a neural execution engine (NEE). These neural components can be plugged into the execution of more complex algorithms, including selection_sort, merge_sort, shortest_path, minimum_spanning_tree. Specifically, a program of the corresponding algorithm is already provided, and all other code lines are symbolically executed as usual, and only the abovementioned sub-routines are replaced with the neural execution engine. Similar to existing work on neural algorithm induction and neural program synthesis, their goal is to achieve out-of-distribution generalization, e.g., when the test samples are longer than training samples for sorting, and when the test graphs are larger than training graphs. They observe that standard Transformer architectures do not generalize in these scenarios, even if the model is trained with strong supervision, i.e., intermediate execution states. Therefore, they propose to integrate a mask prediction component, which restricts the model to only look at a subset of the input for the next computation step. Empirically, they show that this variant achieves 100% generalization performance in many cases when the test samples are larger than training ones.

Strengths: Out-of-distribution generalization is generally an important topic, and it is especially critical for algorithm learning. The model achieves strong results for learning to execute several algorithms.

Weaknesses: In general, I think the technical novelty of this work is limited. In particular, they claim that an additional mask prediction component is necessary to achieve generalization. My understanding is that the training supervision of NEE includes the desired mask at each execution step, which corresponds to the data pointers. However, it is unclear whether the training supervision of the baseline Transformer also includes the ground truth masks, or it only includes the output value at each step. Basically, I want to know whether the improvement comes from the more fine-grained supervision or the architectural design. Some experimental results are not explained clearly. Assuming that the learned sub-routines are perfectly correct, the results should always be 100% regardless of the input size. However, in Table 1, the accuracy of the shortest path problems is not always 100% when the number of nodes exceeds 90. Meanwhile, in Table 5 of the supplementary material, with different graph types, the accuracy of minimum spanning tree starts to drop when the graph size is 50. Is there any explanation for the errors? In Table 2, they present the results when not all numbers during inference have appeared in training set. By looking at Figure 5 (c), I feel that the training numbers are randomly selected from 0~255. What if the unseen test numbers are always larger than training numbers, e.g., the model is trained on numbers 0~63, and testing on numbers >=64? It is hard to understand Figure 4. What are the meanings of x and y axes? UPDATE: I thank the authors for clarifying my questions. It is interesting to see that learning to construct Transformer masks significantly improves the generalization to out-of-distribution test samples. After discussing with other reviewers, I do agree that this paper presents an interesting technique to achieve generalization for learning some basic algorithms. However, since there has been existing work showing that with strong supervision on execution traces, neural networks can achieve 100% out-of-distribution generalization accuracy on learning simple algorithms (i.e., recursive NPI), and a line of subsequent work shows that we can train the model with weaker supervision, I don't think the technical novelty of this paper is significant. Therefore, I still keep my initial score.

Correctness: The technique itself looks correct. However, the training details are unclear. It would be helpful if the authors can discuss the training supervision with more details, for both NEE and the baseline Transformers. Meanwhile, I feel that their main empirical contribution is on learning to execute the subroutines, rather than solving the composed algorithms, given that: (1) the algorithm sketch is already provided without any learning required; and (2) the training of each NEE component is separated.

Clarity: The paper is generally well-written, though some details are unclear.

Relation to Prior Work: The paper provides a good discussion of related work.

Reproducibility: Yes

Additional Feedback:

[Author Response · NeurIPS 2020]

We thank the reviewers for detailed and constructive reviews and respond to each reviewer's questions:

**R1: Utility of Strong Generalization:** Generalization and strong generalization on out-of-distribution data (co-variate shift) are fundamentally unsolved problems in both machine learning and specifically program synthesis/neural execution. Our insight that faithfully learned feedback masks helps strong generalization of transformers in a difficult domain is an important contribution to understanding why strong generalization occurs.

**Importance of Strong Supervision:** Our primary objective with this paper is to establish whether strong generalization can be achieved (to a reasonable degree) in a mechanistic sense. This is effectively a form of imitation learning. Followup work would focus on relaxing this constraint, but we believe that this is beyond the scope of a single paper. Other papers in the field have taken a similar path, for example [21] explores program synthesis via strong supervision and Pierrot et al., 2019 build on this with an RL solution called AlphaNPI that uses weak supervision.

**[29] Comparison:** Code for [29] is not available, making faithful reproduction difficult. NEE & [29] are based on different mechanisms. At each step, [29] requires supervision on every node and edge, while NEE instead supervises on the mask and nodes. We tried to match their setting (using the same graph types) and define our accuracy using a stricter metric (an exactly correct whole-graph result instead of next-node accuracy). For MST, [29]'s next-node accuracy at 100 node graphs is $41.37\%$, implying lower accuracy with the strict whole-graph correct metric vs close to $100\%$ for NEE. There are evalua-

(a) Before training       (b) After training

Figure-R 1: PCA projections on number embeddings

tion differences ([29] trains using 20-node graphs, NEE trains on 8-node graphs), but we feel our evaluation vs. [29] is fair. [29] suffers from attention degradation which they try to counteract with max aggregation and entropy penalties.

**Binary Ablation:** We provide a binary ablation in the Appendix: (Fig.7, all_modifications variant V.S. all_modifications_one_hot_emb variant). **Figure-5 Insight:** We don't believe that it's expected for close numbers to necessarily show structure due to PCA. Figure-R 1 shows the embedding projections before and after training.

**R2: Motivation:** Briefly, we establish that, even with strong supervision, transformers can't generalize on simple algorithmic tasks. We propose a learned conditional masking mechanism as a fundamental technique to improve this. **Results:** We can't say that performance will be perfect in general, but we observed this behavior in our experiments. On the MST task, we do notice some degradation over large test graphs, but the degradation is graceful. **Figure-5:** We'll work on numerical analysis. Our goal was to establish that a) these embeddings have structure, b) these structures differ meaningfully across tasks, and c) non-observed numbers are embedded correctly into their appropriate spots.

**R3: Baseline Transformer Masks:** We apologize for not being clear here. Yes, both NEE and the baseline transformer have the same strong supervision granularity (output value & correct mask). During training, the baseline transformer is also given the ground truth masks (Figure 3). During testing, masks are not provided for both models and the baseline's attention mask consequently generalizes poorly (Figure 4). NEE's improvement is due to the learned mask and architectural design. A plain transformer without mask modifications at both training and testing times performs poorly (which is why we started working on NEE). We'll include this ablation in the next version of the paper.

**Figure 4:** The element in the $(x, y)$ grid indicates the strength of the attention score between the $x$th number and the $y$th number. Figure-4 shows that the attention in the first 8 rows of the baseline transformer (below training data length, attention between the smallest 8 numbers) is clear, but out of distribution attention (beyond 8 numbers) grows fuzzy, resulting in incorrect predictions.

**Performance:** Yes, we find that using subroutines that strongly generalize, the resultant composition also strongly generalizes. In Table-1 we evaluate a challenging setting (training on 8-nodes, testing on 100). In two cases for shortest path, accuracy isn't exactly $100\%$, but $99.99\%$ and $99.98\%$, which we don't consider significant deviations.

Figure-R 2: Histograms of data used in MST(Setting 2)

In Table-5, using the learned subroutines we achieve $100\%$ accuracy - scenario 1 (S1, first two table rows). You refer to the second two rows (S2), where we train using only Erdős-Rényi random graphs but test with a mix of 4 kinds of graphs (Section A.3, Appendix). In this case, the test distribution drifts from the training distribution (Figure-R. 2), and performance suffers, although the resultant accuracy is still much higher than [29].

**Holding Out Numbers:** Yes, the numbers are randomly selected. You are correct that if a bit is not toggled (i.e. bit 6 for 64 in your example), the model would not be able to generalize, but this motivates future work to create numerical basis that extrapolate across unseen bits, whereas our focus here is on interpolation.

[Meta-Review · NeurIPS 2020]

After reading each others reviews and discussing the author rebuttal, opinions on this submission sit around the borderline. The hesitation towards acceptance largely comes from a confusion around the key motivations of the work, the amount of important details residing in the supplement, and uncertainty around the relationship to prior work in program synthesis. The work focus on strong generalization in neural networks trained to perform algorithmic reasoning. The experiments focus on generalization in a fairly narrow domain -- algorithmic subroutines such as finding the minimum of a list, merging two sorted lists, taking a sum, etc. And the type of generalization examined is concerned with extending the length of the input list by scaling up the associated problem (sorting, MST, shortest path) and generalizing to never-before seen numbers or number combinations. The paper demonstrates poor generalization along these dimensions for standard transformers under identical input/output schemes. A modification to the architecture and number embedding is suggested and shown to generalize to significantly longer inputs and to new numbers. The supplement provides more detailed ablation of these changes for the sorting setting. The main experiments section lacks the standard Old vs Ours comparison between the proposed NEE and a vanilla transformer; however, the previous section establishes the vanilla model's relative weakness. Though it does seem that the setting there for sorting is slightly different than the find_mind used in the final experiments. That said, direct comparisons with vanilla transformers throughout would make the paper's presentation stronger and provide additional data points in confirmation with the prior result. Overall, the work suffers perhaps in accessibility and presentation more than in conception or execution. Leaving many details for the supplement. Not providing clear comparison with the baseline. Not contrasting with prior work [3] that is closely related in motivation. The paper is likely to be of interest to a group of researchers; however, it could be made stronger with some revision to improve the presentation.